# Antifouling PVC Catheters by Gamma Radiation-Induced Zwitterionic Polymer Grafting

**DOI:** 10.3390/polym14061185

**Published:** 2022-03-16

**Authors:** Lorena Duarte-Peña, Emilio Bucio

**Affiliations:** Departamento de Química de Radiaciones y Radioquímica, Instituto de Ciencias Nucleares, Universidad Nacional Autónoma de México, Circuito Exterior, Ciudad Universitaria, Ciudad de México 04510, Mexico

**Keywords:** antifouling, gamma radiation, grafting polymerization, pH-responsive, protein adsorption, zwitterionic materials

## Abstract

In medical environments, polymeric surfaces tend to become contaminated, hindering the treatment and recovery from diseases. Biofouling-resistant materials, such as zwitterionic polymers, may mitigate this problem. In this work, the modification of PVC catheters with a binary graft of 4-vinylpyridine (4VP) and sulfobetaine methacrylate (SBMA) by the oxidative pre-irradiation method is proposed to develop pH-responsive catheters with an antifouling capacity. The ionizing radiation allowed us to overcome limitations in the synthesis associated with the monomer characteristics. In addition, the grafted materials showed a considerable increase in their hydrophilic character and antifouling capacity, significantly decreasing the protein adsorption compared to the unmodified catheters. These materials have potential for the development of a combined antimicrobial and antifouling capabilities system to enhance prophylactic activity or even to help treat infections.

## 1. Introduction

Advancement in medicine promotes technological and scientific development since it always seeks to improve the conditions to maintaining physical well-being. In the biomedical field, the use of polymeric materials has boomed in recent decades in the manufacture of tools that are commonly employed in medical treatments (for instance, orthopedic implants, catheters, infusion and dialysis lines, vascular stents, and sutures) due to the versatility of their physical and chemical properties, such as resistance to chemical attack, biocompatibility, low weight, availability, and ease of manufacturing, which allow for adaptability and compatibility with living tissues. 

However, polymeric materials used in medical environments tend to suffer biological contamination with the proliferation of microorganisms as biofilms, leading to infections resistant to conventional antibiotics [1,2] and colloidal contamination that can cause thrombi by blood coagulation onto implanted surfaces [3]. Both contaminations represent a danger for medical procedures, and their presence increases mortality and morbidity rates and decreases the useful life of the devices [4,5].

Research regarding biomedical polymeric materials has focused on biofouling-resistant surfaces development through active coatings or by the surface modification of conventional polymers. There are two categories of materials that prevent microbial infection in medical tools, antimicrobial and antifouling materials [6]. The former inhibit or kill microorganisms because they have antimicrobial activity, which is achieved by some active agent release or by the presence of one on the surface, for example, the covalent bond of quaternary ammonium salts or biocide molecules; and the second prevents microorganism adhesion through hydration layers or lowering the surface energy since a low surface energy leads to high interfacial tension between the liquid and the substrate, which minimizes adhesion [7].

Over the years, the development of antifouling polymers has gone through several basic models. Three generations of antifouling polymers are generally recognized: polymers based on 2-hydroxyethyl methacrylate (HEMA) [8,9], polymers based on polyethylene glycol (PEG) [10,11], and zwitterionic polymers [12]. Zwitterionic polymers are neutral macromolecules formed by monomers with an ionic par (cation–anion) in their structure, and thus these polymers have a large number of ions along their chain, which allows them easily interact with water as they are very hydrophilic [13,14]. 

Their fouling resistance occurs due to the formation of an electrostatically stabilized hydration layer. In this case, hydration occurs through ion–dipole interactions (stronger than van der Waals interactions): the ions over the surface generate repulsive electrical double layer (EDL) forces, which produce a high-energy barrier that prevents surface contamination [15,16]. The poly (sulfobetaine methacrylate) (polySBMA) is a zwitterionic polymer that has demonstrated its antifouling capacity in hydrogel and coating production but presents difficulties for covalently bonding to surfaces [17,18,19,20,21].

On the other hand, stimuli-responsive polymers are often used for antimicrobial system manufacture. These polymers suffer structural changes depending on the medium conditions, such as temperature, pH, ionic strength, and light, which allows them to act as active agent delivery systems [22,23]. Poly (4-vinylpyridine) is known for its pH-responsive properties due to the protonation of the aromatic amine, and thus it has been applied in drug-carrying research. In addition, this polymer demonstrated thermo-responsiveness under certain conditions [24].

The first step in colloidal and biological contamination is protein adsorption on the surface. When a biological fluid comes into contact with a solid interface, proteins are the first to adhere, forming a conditioning film over the surface [25,26]. Therefore, surfaces that avoid protein adsorption may be an effective antifouling method. Protein adsorption on the material is a phenomenon influenced by the external parameters, the kind of protein, and the surface characteristics, which may be affected by several processes, such as structural rearrangements, changes in the surface affinity, size exclusion effects, and protein surface aggregation [27].

This work proposes the modification of PVC catheters with a two-step binary graft of 4-vinylpyridine (4VP) and SBMA ([PVC-*g*-4VP]-*g*-SBMA), using gamma radiation to obtain a pH-responsive material with antifouling characteristics that were tested by measuring the adsorption of bovine serum albumin (BSA) at physiological-like conditions. The main aim was the synthesis of SBMA grafting by gamma radiation in order to study the effects of the different reaction factors over the grafting percentage. [PVC-*g*-4VP]-g-SBMA systems have the potential to be a combined antimicrobial system since they could act as an active agent release system in addition to having antifouling properties.

## 2. Materials and Methods

### 2.1. Materials

PVC catheters (diameter 3 mm and thickness 0.5 mm) were from Biçakcilar (Istanbul, Turkey). 4VP (95%), and SBMA monomers were purchased from Aldrich Chemical, Saint Louis, MO, USA. Methanol was purchased from J. T. Baker (Puebla, Mexico); 4VP was purified by vacuum distillation, and SBMA was used as received. Sodium chloride (NaCl), potassium chloride (KCl), sodium phosphate dibasic (NaH_2_PO_4_), and potassium phosphate monobasic (KHPO_4_) were also from Aldrich Chemical, Saint Louis, MO, USA. Distillate water was used in all the assays. The gamma-ray source was a ^60^Co Gammabeam 651-PT proportioned by the Nuclear Science Institute at National University of México (ICN-UNAM).

### 2.2. VP Graft on PVC Catheters (PVC-g-4VP)

4VP was grafted onto PVC following the process reported in a previous study [28]. The modification was performed using the direct irradiation method, varying the applied doses (10, 15, and 20 kGy) in order to standardize the graft percentages obtained, and the used conditions were a concentration of 40% (*v/v*) 4VP in methanol: water mixture with 30% MeOH, vol.%. The graft percentage was calculated with Equation (1), where W_f_ is the grafted sample weight (g) and W_i_ is the initial weight of the sample (g).
(1)$$\mathrm{Graft}\;\left(\%\right)=\frac{\left(W_{f}{-W}_{i}\right)\;\times \;100}{W_{i}}$$

### 2.3. SBMA Graft on PVC-g-4VP Catheters [(PVC-g-4VP)-g-SBMA)]

SBMA grafting was performed on PVC-*g*-4VP catheters using the oxidative pre-irradiation method. For the synthesis, dry PVC-g-4VP catheters were weighed and placed in glass ampoules, which were then irradiated in the presence of air. A SMBA solution in MeOH: H_2_O was added to the irradiated samples, the oxygen in the solution was removed by air displacement with argon bubbling for 10 min, and then the ampoules were sealed and kept at 5 °C for 4 h to then heat in a water bath. Finally, the samples were removed from the ampoules and washed, once with a saturated NaCl solution for 2 h, twice with distilled water for 3 h, and once with ethanol; after which they dried under vacuum oven for 12 h at 30 °C, and the percentage of grafting was calculated following Equation (1). Figure 1 shows the procedure.

The effect of the different reaction conditions on the SBMA graft percentage was studied, for which the conditions were varied: the 4VP percentage grafted (4, 12, and 21%), doses (1, 2, 5, 10, 15, and 20 kGy), dose rate (5.4 and 13 kGy/h), SBMA concentration (0.05, 0.1, 0.15, 0.2, 0.25, 0.5, 0.75, and 1 M), MeOH percentage in the MeOH:H_2_O mix used as dissolvent (0, 25, 50, 75, and 90%), reaction temperature (40, 50, 60, and 70 °C), and reaction time (6, 8, 12, 18, and 24 h).

### 2.4. FT-IR and Thermal Characterization

FT-IR spectroscopy analyses for grafting identification were performed on a Perkin-Elmer Spectrum 100 spectrophotometer (Perkin Elmer Cetus Instruments, Norwalk, CT, USA) with 16 scans, in the ATR module, in the range of 4000 to 650 cm^−1^. In addition, the thermal behavior of the materials was monitored using thermogravimetric analysis (TGA) under a nitrogen atmosphere from 30 to 700 °C at a heating rate of 10 °C/min in a TGA Q50 (TA Instruments, New Castle, DE, USA).

### 2.5. Wettability

#### 2.5.1. Dynamic Contact Angle

The wettability degree was analyzed by the sessile drop method, using a goniometer DSA 100 Krüss GmbH, Hamburg, Germany. The dynamic contact angle was measured at 0, 2, and 5 min of interaction; all measurements were performed four times.

#### 2.5.2. Swelling

The dry sample was weighed and placed in a glass with the solvent at a controlled temperature; distilled water at 25 °C and phosphate buffer (PBS) at 37 °C. The sample was removed from the beaker and weighed after removing excess solvent; this process was performed every 5 min for the first fifteen minutes and then at 0.5, 1, 2, 4, 6, and 12 h. The swelling percentage was determined using Equation (2), where W_1_ is the dry sample weight (g) and W_2_ is the swollen sample weight (g); each measurement was made in triplicate.
(2)$$\mathrm{Swelling}\;\left(\%\right)=\frac{\left(W_{2}{-W}_{1}\right)\;\times \;100}{W_{1}}$$

### 2.6. pH and Thermo Responsiveness

For the pH responsiveness test, buffer solutions of different pH (2, 3, 4, 5, 6, 8, 10, and 12) were prepared. The dry sample was weighed and swelled at 25 °C for 2 h in each pH solution. The swelling percentage was calculated with Equation (2) The critical pH and the pH-sensitivity determined the pH responsiveness; the former was estimated as the inflection point in the swelling (%) vs. pH curve; and the latter was calculated with the swelling ratios below and above the critical pH.

On the other hand, for thermo-responsiveness, dry samples were weighed and swelled for 2 h at different temperatures (5, 25, 37, and 50 °C) in water and buffer solutions with pH 4 and 10. The swelling percentage was calculated with Equation (2).

### 2.7. Protein Adsoption

Around 80 mg of sample were placed in a PBS buffer solution at 37 °C for 2 h, and then the sample was incubated for 2 h in bovine serum albumin (BSA) protein solution in PBS with a concentration of 30 mg/mL, at 37 °C. At the end of this time, the material was washed three times with PBS, and the adsorbed protein was extracted and later quantified using a spectroscopic method.

#### 2.7.1. Protein Extraction

To a sample that was previously incubated in BSA, we added 600 µL of 1% wt. sodium dodecyl sulfate (SDS) solution, then stirred at 130 rpm for 20 min and sonicated for 10 min, and finally vortexed for 30 s; this process was performed three times.

#### 2.7.2. Protein Quantification

The bicinchoninic acid (BCA) method was used for quantification [29,30]. We added 1 mL of the extracted protein sample and 2 mL of working solution in a glass vial; the mixture was gently shaken and allowed to react at 60 °C for 30 min. After this time, the sample was cooled to room temperature and measured by UV-Vis spectroscopy at 556 nm. The quantification was performed using a calibration curve. In all cases, the reference was a 1% wt. SDS solution, to which the process described in Section 2.7.1 was applied. The calibration curve and the preparation of the working solution are in the attached material.

## 3. Results

### 3.1. PVC-g-4VP Synthesis

The 4VP grafting was obtained using the direct irradiation method; a study of the reaction conditions on the grafting percentage was reported in previous research [28]. The interaction between PVC and gamma radiation has been widely studied with the conclusion that two primary reactions are produced: ionization and excitation, which trigger many secondary reactions. The main reactions that occur with PVC radiation exposure are shown in Figure 2; radical (I) is the main product due to the system stability [31]. In the direct irradiation method, the monomer interacts directly with the radicals on the surface and produces grafting.

In this work, we investigated the standardization of reaction conditions to obtain 4VP grafting percentages between 2 and 20%, which allowed studying the effect of this grafting percentage on the modification with SBMA and the properties of the material. Figure 3 shows the results obtained, with relative standard deviations of approximately 15%, which are acceptable due to the complexity of the methodology.

### 3.2. SBMA Graft on PVC-g-4VP Catheters [(PVC-g-4VP)-g-SBMA)]

Figure 4 shows a schema of the SBMA grafting onto PVC-*g*-4VP catheters; this synthesis was performed using the oxidative pre-irradiation method, where the dose, dose rate, monomer concentration, solvent, temperature, and reaction time may significantly affect the graft percentage. The 4VP grafting was used to endow the PVC surface with a hydrophilic character, which allowed its interaction with monomeric SBMA. 

The SBMA could not graft directly onto the PVC because the PVC hydrophobic nature avoided interaction with the SBMA, which is stabilized by internal electrostatic forces, which favors homopolymer formation. Figure 5a shows the SBMA grafting percentages obtained for PVC catheter grafted with different 4VP amounts (4, 12, and 21%), where we did not observe a relevant change, and thus we decided to work with PVC-*g*-4VP (4%) catheters in the subsequent analyses.

The absorbed dose is an average of the ionizing radiation energy that the material receives. This parameter determines the number of radicals formed, directly influencing the grafting percentage [32]. In the oxidative pre-irradiation method, oxygen interacts with the radicals produced by gamma radiation (Figure 2), forming peroxide and hydroperoxide bonds that later break with temperature and give rise to a chain reaction to form the graft. The 4VP has a higher radiation resistance than PVC owing to its aromaticity, which suggests that, even in the modified PVC-*g*-4VP catheter, mainly radical (I) will be formed. The small influence of the 4VP grafting percentage in the SMBA grafting is another indication that radicals were mainly formed on the PVC (Figure 4).

Figure 5b shows the SBMA grafting behavior on a sample of PVC-*g*-4VP (4%) when irradiated with different doses—0.5, 1, 2, 5, 10, 15, and 20 kGy. A constant behavior was observed, which indicated that, in this dose range, the dose did not have a significant effect on the SBMA graft percentage.

The dose rate is the dose absorbed by a sample per unit of time. This parameter affects the radical formation rate at the beginning of the reaction, which is very relevant in the direct irradiation method, in which simultaneous formation of the homopolymer and the graft takes place [33]. To determine the effect associated with the dose rate, SBMA was grafted onto PVC-*g*-4VP (4%), varying only the dose rate at 5.4 and 13 kGy/h. Figure 5c compares the SBMA grafting percentages for these dose rates; the results did not show a significant difference between them, which was confirmed by the t-significance contrast test [34]; see the Appendix A. The results indicated that the dose rate is not a relevant parameter in the SBMA grafting; thus, the grafting will not be affected by the change in the dose rate due to ^60^Co source attrition.

The monomer concentration influences the diffusion of the monomer on the surface to be grafted and the homopolymer formation since, at high concentrations, the Trammsdorff effect occurs, which favors gel formation instead of grafting. In the case of zwitterionic monomers, their agglomeration occurs even at relatively low concentrations. The SBMA concentration effect on the synthesis of (PVC-*g*-4VP)-*g*-SBMA was studied by grafting SBMA onto PVC-*g*-4VP (4%), Figure 5d shows the results, which presented a bell behavior with the maximum grafting at a concentration of 0.5 M. The 0.25 M SBMA concentration was chosen for analysis since it allows a wide range of grafting.

The solvent was essential to achieve the grafting of SBMA on the PVC-*g*-4VP (4%) catheters since it must allow for the dissolution of the monomer and for the monomer–surface interaction. Figure 5e shows a graph of the SBMA grafting percentage as a function of the methanol percentage in the solvent that consisted of a mixture of methanol and water. When pure water or low percentages of MeOH were used as the solvent, SBMA grafting did not occur, which was attributed to the monomer stabilization by solvation, which prevents interaction with the surface. The highest MeOH content mixtures (75 and 90%) showed the highest percentage of grafting; however, with 90% MeOH, a complete dissolution of the SBMA was not achieved.

The temperature triggers the homolytic cleavage of the peroxides and hydroperoxides on the surface, leading to grafting formation. Figure 5f shows the SBMA grafting percentages obtained at different temperatures; temperatures below 60 °C showed few differences in the modification degree, while at 70 °C, the grafting percentage significantly increased. Finally, the effect of the reaction time on the SBMA graft percentage is shown in Figure 5g with a proportional increase between the two parameters.

### 3.3. FT-IR and Thermal Characterization

The grafted catheters were analyzed by infrared spectroscopy (ATR-FTIR) for each grafted polymer characteristic functional group identification, Figure 6. The FT-IR spectrum of the pristine PVC catheter showed bands at 2992 cm^−1^ that correspond to C-H stretching of -CH_2_ groups and at 1267 cm^−1^ of the C-H bending, and bands at 1745 and 1459 cm^−1^ characteristic of the plasticizer. 

The PVC-*g*-4VP spectrum, apart from the previous bands, showed C=C stretching signals at 1597 and 1557 cm^−1^ and a band at 830 cm^−1^ from C-H bending out of the plane of the aromatic amine. Finally, the (PVC-*g*-4VP)-*g*-SBMA spectrum showed, in addition to all the above bands, bands at 1039 and 1265 cm^−1^ corresponding to the symmetric and asymmetric stretching of the sulfonate group (-SO_3_^−^), a band from the stretching of the carboxylic group at 1736 cm^−1^, and at 1440 cm^−1^ from the quaternary amine [35]. Higher SBMA grafted percentages presented the same bands but were more pronounced.

TGA thermic analysis showed that the PVC matrix determines the thermal stability of PVC-g-4VP and (PVC-*g*-4VP)-*g*-SBMA catheters since they lost 10% of weight at approximately the same temperature (238 ± 2 °C). The 4VP and SBMA homopolymers showed higher stability with losses of 10% in weight at 362 and 310 °C, respectively. (PVC-*g*-4VP)-*g*-SBMA catheters presented three decomposition temperatures. The first was at 266 °C due to PVC dehydrochlorination [36], which occurred at temperatures that were close in both PVC and PVC-*g*-4VP; the second was at 328 °C, which agrees with the first decomposition of polySBMA at 341 °C (due to the degradation of the quaternary anima [37]) and the second decomposition of PVC at 288 °C; and the third was at 451 °C, which corresponds to the carbon chain decomposition. Figure 7 shows the thermograms of each sample.

### 3.4. Wettability

#### 3.4.1. Dynamic Contact Angle

The contact angle allows measuring the interaction between a liquid and a solid in a three-phase system (solid, liquid, and gas), depending on the angle formed by the liquid into contact with the solid surface. When a surface strongly attracts liquid molecules, the liquid drop spreads over it, leading to contact angles of less than 90°; on the contrary, if the cohesive forces are higher, the angle will have a value greater than 90° [38]. Figure 8 shows the contact angles with water for the samples of (PVC-*g*-4VP)-*g*-SBMA and compared with PVC-*g*-4VP and PVC. 

The pristine PVC is a hydrophobic material with a contact angle greater than 90°, around 100°; 4VP modified PVC angles decrease proportionally to the modification to 55° for 12 and 65° for 4%; finally, the second graft with SBMA showed the angles close to 30° for 10% graft and 20° for 23% graft at 5 min of interaction. The PVC and PVC-*g*-4VP contact angles remained constant, while the SBMA grafted catheters showed contact angles that decreased with time until the water drop adsorbed at 10 min.

#### 3.4.2. Swelling

Swelling in water is an indication of the hydrophilicity of the material. Figure 9a,b shows the swelling profiles for the (PVC-*g*-4VP)-*g*-SBMA samples based on PVC-*g*-4VP (4%) and PVC-*g*-4VP (12%), respectively, and different contents of SBMA and their comparison with the pristine PVC swelling profile. In both cases, unlike pristine PVC, the modified materials had a greater water affinity and, therefore, presented a higher swelling percentage. The hydrophilicity was significantly higher for the SBMA grafted samples, where the swelling was twice that shown by 4VP-only grafted catheters. Comparing the catheters with the same SBMA content but different 4VP contents (4 and 12%), there were no relevant differences among the swellings, which indicates that the SBMA percentage dominated the interaction water/material. For all samples, the limit of the swelling time was 2 h.

On the other hand, Figure 9c,d shows the swelling profiles in PBS buffer solution for [PVC-*g*-4VP (4%)]-*g*-SBMA and [(PVC-*g*-4VP (12%)]-*g*-SBMA catheters. PBS is a solution based on phosphate salts, which allows a constant pH at 7.4 and ionic strength of 0.14 M, which are conditions similar to human blood. PBS solution is widely used to evaluate materials with biomedical applications. The assays were performed at 37 °C to analyze the material behavior in conditions that are similar to physiological. As in the case of water, a significant increase in the swelling percentage for SBMA grafted materials was observed, and the SBMA percentage determined the material behavior. 

Figure 9c shows that the [PVC-*g*-4VP (4%)]-*g*-SBMA samples with 13 and 23% of SBMA grafting presented a singular behavior, with a maximum swelling at 30 min that later stabilized at 2 h, which was due to an ionic effect; in the case of samples with 12% 4VP, this percentage allowed higher stabilization. Finally, no significant difference was observed when comparing the swelling in water and PBS.

### 3.5. pH Responsiveness

The pH responsiveness is a characteristic of polymers whose structure contains ionizable groups, which present charges depending on the pH of the medium; when these groups ionize, a repulsion force is generated, and the materials attract more water to stabilize, thus, increasing the swelling. In this case, poly(4VP) is a pH-responsive polymer whose critical pH is around 5.4 [39,40]; 4VP protonates at a pH less than the critical pH, and the swelling increases. Figure 10a,b shows the swelling curves as a function of pH for the PVC, PVC-*g*-4VP, and PVC-*g*-4VP-g-SBMA catheters with different graft percentages. 

All 4VP grafting materials showed pH-responsive behavior with a critical pH of around 7. The test showed that SBMA grafting did not significantly alter the critical pH of the materials, and the 4VP that was covalently bound to PVC displaced its critical pH, varying from 6.6 to 7 in the different samples. In addition, the pH sensitivity remained constant for all the materials, in the range of 1.43 ± 0.08. However, SBMA grafting considerably increased the swelling concerning the base PVC-*g*-4VP.

The temperature responsiveness test showed that none of the materials had sensitivity to the temperature in water or buffer solution at pH 4. On the other hand, the PVC grafted with 4 and 12% of 4VP showed a decrease in swelling when the temperature changed from 5 to 37 °C at pH 10; this behavior corresponds to materials with a low critical solution temperature (LCST) [24]; Figure 11 shows the results. However, grafted catheters (PVC-*g*-4VP)-*g*-SBMA did not present this property due to the SBMA influence.

### 3.6. Protein Adsorption

Figure 12 shows the amount of BSA protein adsorbed on the surface when interacting with a solution that contains the BSA plasma concentration (30 mg/mL). The pristine PVC adsorbed around 5.14 µg BSA/cm^2^; this protein adsorption decreased 35 and 58% for the samples modified with 4VP grafting at 4 and 12%; however, the sample with 21% grafting of 4VP showed an increase of 12% in the protein adsorption. 

On the other hand, the materials modified with 4VP and SBMA showed a significant decrease in the amount of protein adsorbed at 88, 68, 78, and 89% for [PVC-*g*-4VP (4%)]-*g*-SBMA (13%), [PVC-*g*-4VP (4%)]-*g*-SBMA (23%), [PVC-*g*-4VP (12%)]-*g*-SBMA (12%), and [PVC-*g*-4VP (12%)]-*g*-SBMA (25%) catheters, respectively. The results showed that the zwitterionic polymer grafted by this method maintained its antifouling character and provided the modified surfaces with an antifouling capacity for BSA protein, which is a promising result in the development of antimicrobial and hemocompatible surfaces.

## 4. Discussion

PolySBMA has been shown to have high functionality as an antifouling coating. Nevertheless, polySBMA covalent grafting on surfaces is limited since polySBMA is insoluble in virtually all solvents, except for aqueous electrolyte solutions and polar protic solvents, such as trifluoroethanol and hexafluoroisopropanol, which are typically not compatible with the conventional polymerization methods. Gamma radiation is a type of high-energy ionizing radiation, which excites the matter and gives rise to the formation of free radicals. In the oxidative pre-irradiation method, radicals are generated on the surface, which favors grafting. 

This method allowed SBMA grafting onto PVC surfaces in the range of 10 to 50% using methanol:water as the dissolvent. This grafting degree was higher than those obtained by other methods, such as plasma modification [41] or electrochemical surface-initiated atomic transfer radical polymerization [42,43]. In addition, the grafting was conducted at low temperatures and without external agents, such as catalysts or initiators. The grafted materials acquired the antifouling activity of the zwitterionic polymer, reducing protein adsorption on the surface by up to 89%. These materials are promising for the manufacture of catheters with longer useful life and less prone to bacterial adhesion than the currently available.

Figure 13 shows the main characteristics of the synthesized grafted materials: swelling in body conditions (pH 7.4 and T 37 °C), contact angle, pH sensitivity, and protein adhesion. The SBMA grafted materials presented similar characteristics depending on the percentage of grafting, with the materials with approximately 23% SBMA being the ones that showed less protein absorption and higher hydrophilicity. The [PVC-*g*-4VP (4%)]-*g*-SBMA (13%) catheters represent the best option as antifouling materials since they demonstrated less protein absorption and a lower grafted percentage. 

Furthermore, they have the potential to form an antifouling system with an antimicrobial effect due to the release of active agents, such as antibiotics or antiseptics, since they showed a high pH sensitivity, which could favor the release of an active agent in response to changes in the external pH. Currently, most preventing microbial infection products are based on the release of active agents, such as silver and antibiotics, adsorbed or impregnated in the materials [44,45]. Therefore, materials that can achieve a combination of antimicrobial and antifouling properties to reinforce their prophylactic potential or even help in infection treatments are of great interest.

## 5. Conclusions

SBMA was successfully grafted onto PVC-*g*-4VP catheters using the oxidative pre-irradiation method with a grafting range of 10% to 50%. The results showed that the SBMA concentration, MeOH percentage in the solvent, temperature, and reaction time had the most significant influences on the grafting percentage. In addition, the 4VP modification was essential to obtain zwitterionic grafts. Of the synthesized materials, the [PVC-*g*-4VP (4%)]-*g*-SBMA (13%) catheters presented the highest antifouling effects, decreasing the BSA adsorption on the surface by 88% regarding pristine PVC. Additionally, they showed high hydrophilicity, with a contact angle of 28° and PBS swelling of 16% as well as pH-responsive behavior. These materials have the potential for use in systems that are resistant to biofouling and could also act as carriers of antimicrobial agents for biomedical applications.

## Figures and Tables

**Figure 1 polymers-14-01185-f001:**
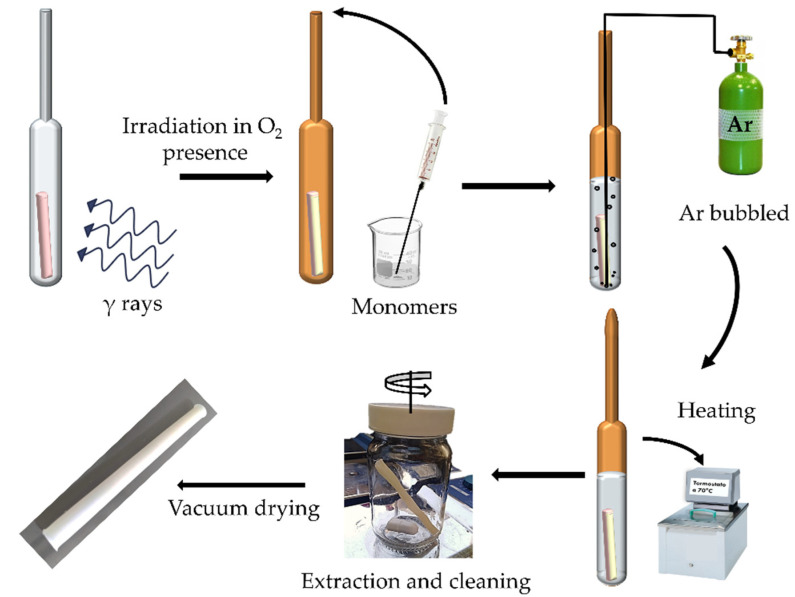
Experimental schematization of oxidative pre-irradiation method.

**Figure 2 polymers-14-01185-f002:**
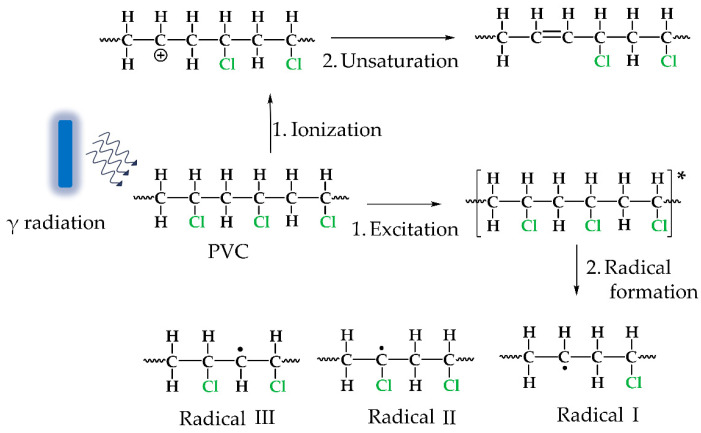
Gamma radiation interaction with PVC, * Excited molecule.

**Figure 3 polymers-14-01185-f003:**
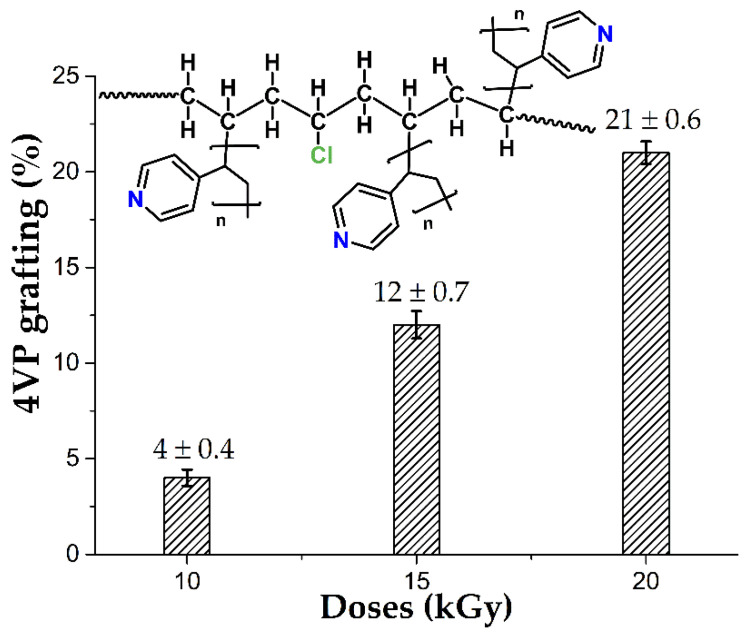
4VP grafting on PVC catheters to change the irradiation doses. Reported: the mean ± standard error of the mean, *n* = 14.

**Figure 4 polymers-14-01185-f004:**
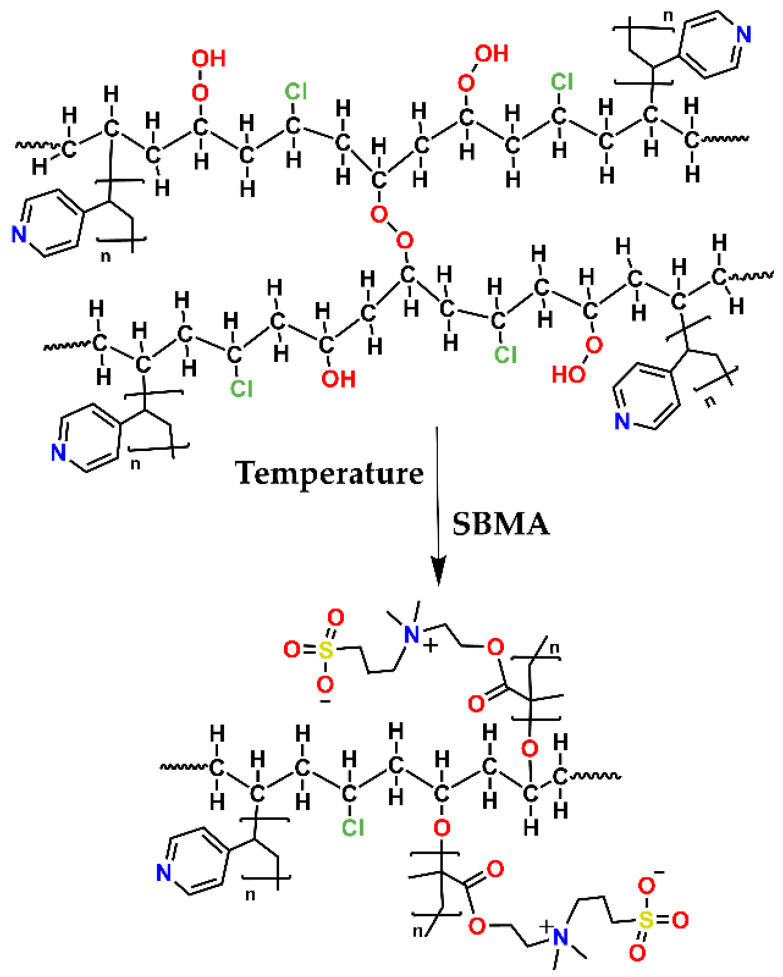
Schematization of the (PVC-g-4VP)-g-SBMA by the oxidative pre-irradiation method.

**Figure 5 polymers-14-01185-f005:**
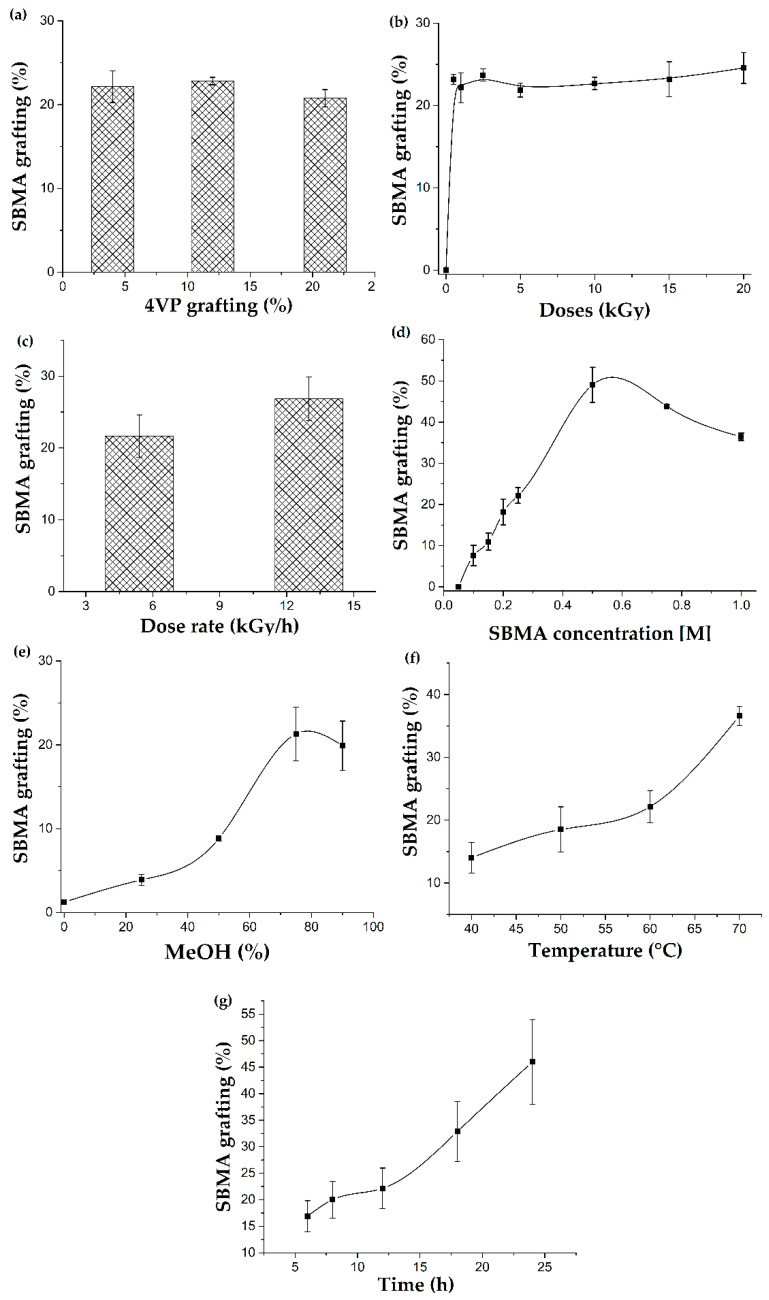
The reaction parameter effects on the SBMA grafting percentage on PVC-g-4VP catheters: (**a**) 4VP grafting percentage (Conditions: 1 kGy, 0.25 M, 75% MeOH, 60 °C, and 12 h); (**b**) Doses (Conditions: PVC-*g*-4VP (4%), 0.25 M, 75% MeOH, 60 °C, and 12 h); (**c**) Dose rate (Conditions: PVC-*g*-4VP (4%), 1 kGy, 0.25 M, 75% MeOH, 60 °C, and 12 h); (**d**) SBMA concentration (Conditions: PVC-*g*-4VP (4%), 1 kGy, 75% MeOH, 60 °C, and 12 h); (**e**) MeOH percentage in the solvent mixture (Conditions: PVC-*g*-4VP (4%), 1 kGy, 0.25 M, 60 °C, and 12 h); (**f**) Temperature (Conditions: PVC-*g*-4VP (4%), 1 kGy, 0.25 M, 75% MeOH, and 12 h); and (**g**) Reaction time (Conditions: PVC-*g*-4VP (4%), 1 kGy, 0.25 M, 75% MeOH, and 60 °C). Reported: the mean ± standard error of the mean, *n* = 3.

**Figure 6 polymers-14-01185-f006:**
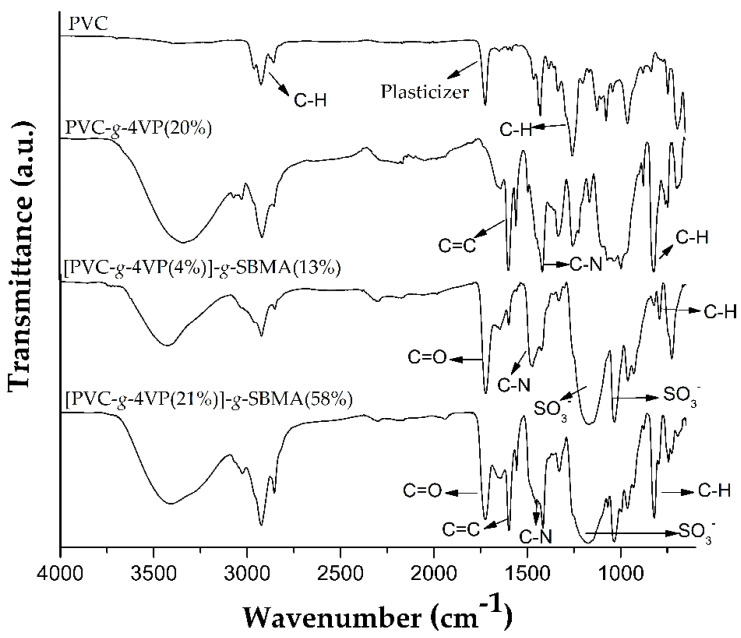
FT-IR spectrums.

**Figure 7 polymers-14-01185-f007:**
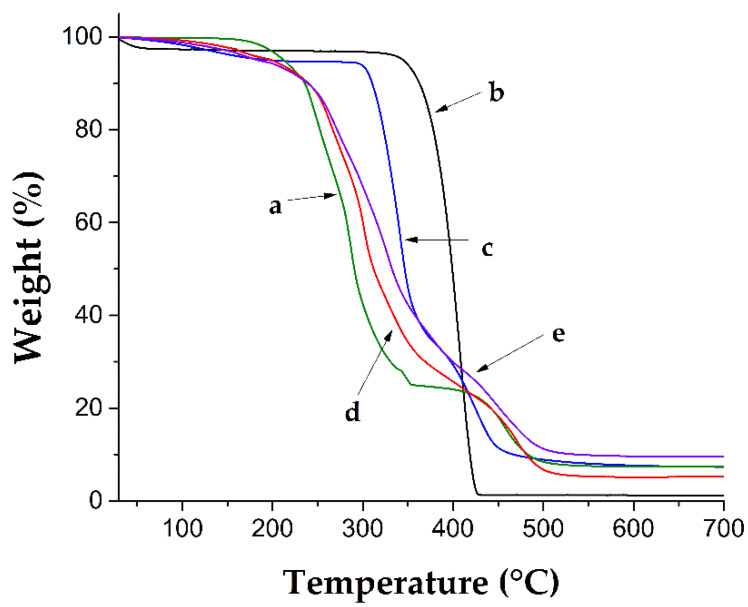
Thermograms (a) PVC, (b) 4VP homopolymer, (c) SBMA homopolymer, (d) PVC-*g*-4VP (12%), and (e) [PVC-*g*-4VP (12%)]-*g*-SBMA (25%).

**Figure 8 polymers-14-01185-f008:**
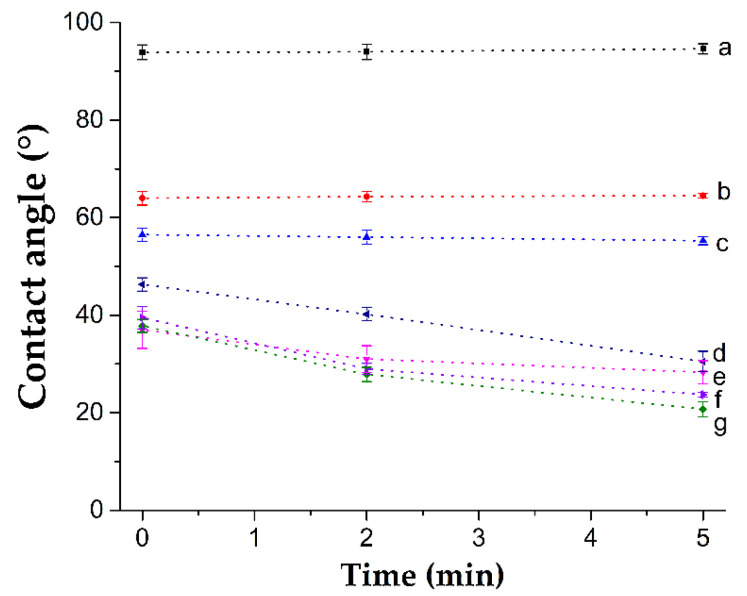
Dynamic contact angles for catheters (a) PVC, (b) PVC-*g*-4VP (4%), (c) PVC-*g*-4VP (12%), (d) [PVC-*g*-4VP (12%)]-*g*-SBMA (12%), (e) [PVC-*g*-4VP (4%)]-*g*-SBMA (13%), (f) [PVC-*g*-4VP (12%)]-*g*-SBMA (25%), and (g) [PVC-*g*-4VP (4%)]-*g*-SBMA (23%). Reported: the mean ± standard error of the mean, *n* = 4.

**Figure 9 polymers-14-01185-f009:**
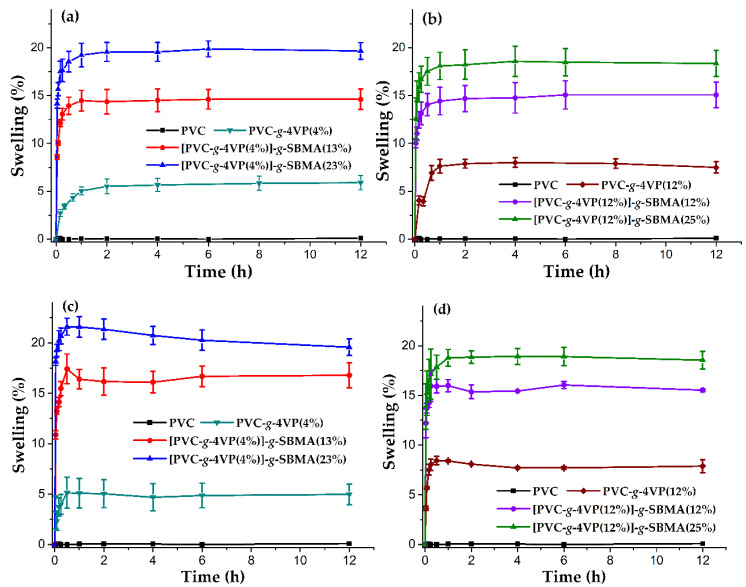
Swelling profiles: (**a**,**b**) in water and (**c**,**d**) in PBS solution. Reported: mean ± standard error of the mean, *n* = 3.

**Figure 10 polymers-14-01185-f010:**
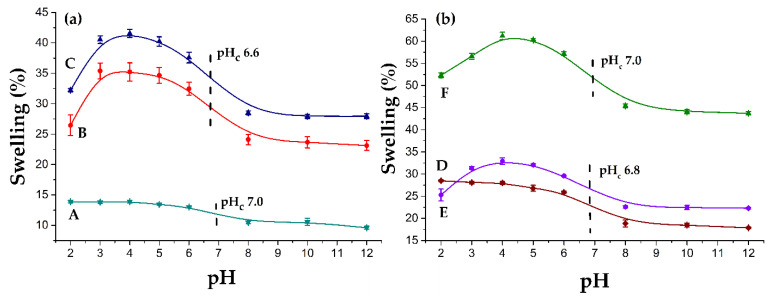
pH responsiveness profiles: (**a**) A: PVC-*g*-4VP (4%), B: [PVC-*g*-4VP (4%)]-*g*-SBMA (13%), and C: [PVC-*g*-4VP (4%)]-*g*-SBMA (23%) and (**b**) D: PVC-*g*-4VP (12%), E: [PVC-*g*-4VP (12%)]-*g*-SBMA (12%), and F: [PVC-*g*-4VP (12%)]-*g*-SBMA (25%). Reported: the mean ± standard error of the mean, *n* = 3.

**Figure 11 polymers-14-01185-f011:**
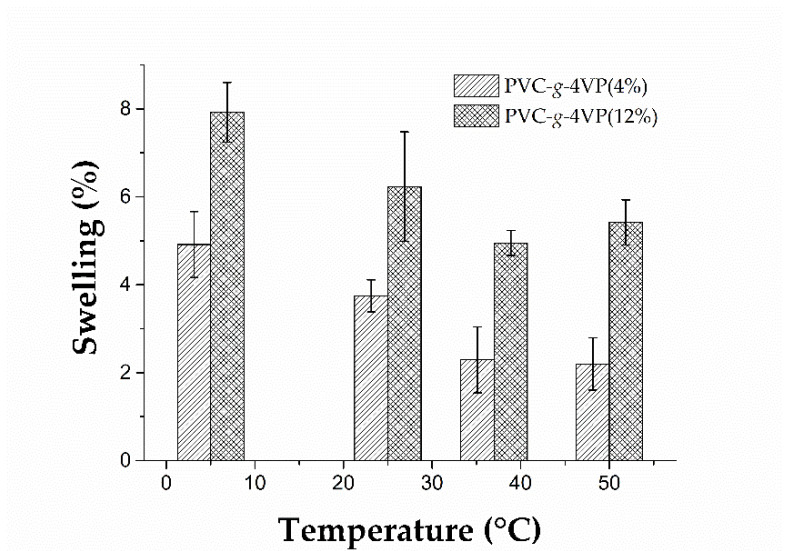
The thermo-responsiveness test. Reported: the mean ± standard error of the mean, *n* = 3.

**Figure 12 polymers-14-01185-f012:**
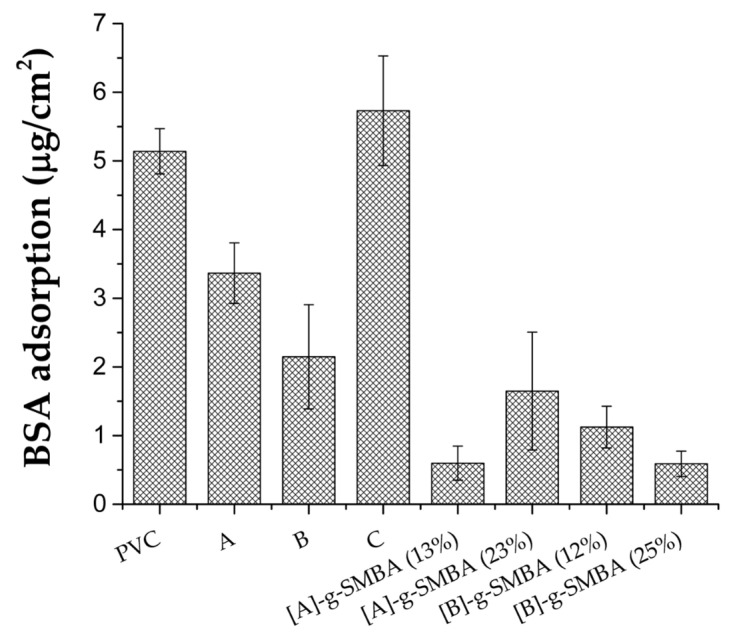
BSA adsorption on PVC catheters grafted with 4VP and SBMA; [A]: PVC-*g*-4VP (4%), [B]: PVC-*g*-4VP (12%), and [C]: PVC-*g*-4VP (21%). Reported: the mean ± standard error of the mean, *n* = 3.

**Figure 13 polymers-14-01185-f013:**
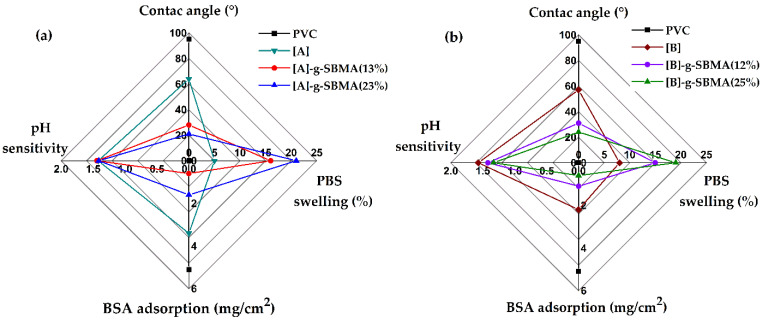
The main properties of SBMA-grafted PVC catheters: (**a**) comparison of the 4% 4VP grafted catheters, [A]: PVC-*g*-4VP (4%) and (**b**) comparison of the 12% 4VP grafted catheters, [B]: PVC-*g*-4VP (12%).

## Data Availability

Not applicable.

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
