# Peer review of "Antifouling PVC Catheters by Gamma Radiation-Induced Zwitterionic Polymer Grafting"

_polymers, 2022, doi:10.3390/polym14061185_

Round 1
Reviewer 1 Report
In work “Antifouling PVC catheters by gamma radiation-induced zwitterionic polymer grafting” is presented interesting results about modification of PVC catheters with a binary graft of 4-vinylpyridine (4VP) and sulfobetaine methacrylate (SBMA) by the oxidative pre-irradiation method is proposed to develop pH-responsiveness catheters with antifouling capacity. Manuscript can be accepted in Polymers mdpi after major revision.
Following important moments should be clear.
- Mentions of the numbers of the figures in the text and subsections are confusing. For example, after subsection 3.1 is subsection 3.3. After subsection 3.6 is subsection 2.7.
- In subsection 3.3 is described mechanism of the grafting of 4VP but omitted information about grafting of the SBMS. I think it will be well to transfer information on 4VP to subsection 3.1 and to provide appropriate discussion including a scheme on the mechanism of the grafting of SBMA to 4VP. Also, it is not clear enough whether are fabricated grafted brushes and what their conformation.
- AFM images modified surfaces and appropriate discussion should be provided.
- Authors describing the method of the protein adsorption noted that “Around 80 mg of sample was placed in a PBS buffer....” but if I undestood rightly the samples are plate surfaces and key role plays surface area but not mass.
- Subsection 3.4 is named as characterization but wettability, swelling and pH-responsivity also are characteristics of the samples.
- What about temperature-responsive properies of the modified surfaces. In some papers information about pH- as well as temperature-responsive properties of P4VP coatings was shown.
- Please add appropriate discussion on the mechanism of the pH-responsivity.
- English should be strongly improved. All manuscript should carefully checked.
- I suggest to cite relevant papers to imrove the quality presented information.
https://doi.org/10.1007/s00396-020-04750-0
https://doi.org/10.1039/C6RA07223B
https://doi.org/10.1039/C3SM51496J
Author Response
- Mentions of the numbers of the figures in the text and subsections are confusing. For example, after subsection 3.1 is subsection 3.3. After subsection 3.6 is subsection 2.7.
Answer: Thanks for the observation, the number of the figures and subsections were carefully revised and changed in the manuscript.
- In subsection 3.3 is described mechanism of the grafting of 4VP but omitted information about grafting of the SBMS. I think it will be well to transfer information on 4VP to subsection 3.1 and to provide appropriate discussion including a scheme on the mechanism of the grafting of SBMA to 4VP. Also, it is not clear enough whether are fabricated grafted brushes and what their conformation.
Answer: The change was performed; the second paragraph of the section was rewritten and a new figure was added.
- AFM images modified surfaces and appropriate discussion should be provided.
Answer: Thank you very much for your suggestion. Unfortunately, it was not possible to acquire the proposed analysis.
- Authors describing the method of the protein adsorption noted that “Around 80 mg of sample was placed in a PBS buffer....” but if I understood rightly the samples are plate surfaces and key role plays surface area but not mass.
Answer: The samples were catheters whose area density depends on the material, we used mass units to be more general in the experimental description, but the results were showed in area units. The following table show the area density for each material:
|
Catheter |
Area density (mg/cm2) |
|
PVC |
42.3 ± 0.48 |
|
PVC-g-4VP(4%) |
40.2 ± 3.79 |
|
PVC-g-4VP(12%) |
43.4 ± 1.56 |
|
[PVC-g-4VP(4%)]-g-SBMA(13%) |
39.1± 3.60 |
|
[PVC-g-4VP(4%)]-g-SBMA(23%) |
35.9 ±1.79 |
|
[PVC-g-4VP(12%)]-g-SBMA(12%) |
37.7 ±3.89 |
|
[PVC-g-4VP(12%)]-g-SBMA(25%) |
37.8 ±2.72 |
- Subsection 3.4 is named as characterization but wettability, swelling and pH-responsivity also are characteristics of the samples.
Answer: The subsection title was changed to “FT-IR and thermal characterization”.
- What about temperature-responsive properties of the modified surfaces. In some papers information about pH- as well as temperature-responsive properties of P4VP coatings was shown.
Answer: Thanks for the suggestion. Temperature-responsiveness test was performed for PVC, PVC-g-4VP, and (PVC-g-4VP)-g-SBMA catheters varying temperature at 5, 25, 37, and 50 °C and using water and buffer solutions with different pH (4 and 10), in which differences in the swelling were measured. The materials did not show sensitivity to temperature in water or buffer solution of pH 4. On the other hand, the PVC grafted with 4 and 12% of 4VP showed a decrease in their swelling when the temperature changed from 5 to 37 °C at pH 10, the following figure shows the results; however, grafted catheters (PVC-g-4VP)-g-SBMA did not present this property. This results were added in section 3.5
Temperature responsiveness for PVC-g-4VP al 4 y 12 %
- Please add appropriate discussion on the mechanism of the pH-responsivity.
Answer: The section 3.5 was modified.
- English should be strongly improved. All manuscript should carefully checked.
Answer: The manuscript was detailly review.
- I suggest to cite relevant papers to improve the quality presented information.
https://doi.org/10.1007/s00396-020-04750-0
https://doi.org/10.1039/C6RA07223B
https://doi.org/10.1039/C3SM51496J
Answer: Thanks for the suggestion, some of the references were added.

Reviewer 2 Report
The authors proposed a surface modification approach to graft antifouling and zwitterionic SBMA on PVC tubes via oxidative gamma pre-irradiation. The coating of SBMA was carefully optimized in terms of dose, dose rate, monomer concentration, solvent, temperature, and reaction time. Properties of the coatings were characterized upon the swelling ratio, grafting density, protein adsorption and thermostability. Over all the work should be interesting to the readers of the journal. However, suggestions below should be considered before accepting the article:
- The calculation of grafting density in the Equation 1 is odd for me. The high grafting density (>20%) seems to me that the coating becomes a important component in the PVC tubes. The portion of the grafted materials is significant and huge as a coating materials.
- In addition to hydrophilicity of 4VP, is any other function of 4VP which affect the deposition/grafting of SBMA? Why not just graft SBMA on PVC directly?
- For the manufacturing, I wonder if you bubble the monomer solution in Ar before adding to react with PVC tubes, (not bubbling during the reaction). Would it be helpful?
- Important reference should be cited:
; Langmuir 35 (5), 1642-1651Langmuir 32 (19), 5019-5028
Author Response
- The calculation of grafting density in the Equation 1 is odd for me. The high grafting density (>20%) seems to me that the coating becomes a important component in the PVC tubes. The portion of the grafted materials is significant and huge as a coating material.
Answer: Equation 1 determines the grafted percentage concerning the original material. We agree that the high percentages of grafting become a relevant component in the PVC tubes. However, in this case, we did not observe a deformation in the PVC catheters until grafted percentages > 30%, due to which we used the materials with lower modification for the characterization.
- In addition to hydrophilicity of 4VP, is any other function of 4VP which affect the deposition/grafting of SBMA? Why not just graft SBMA on PVC directly?
Answer: The hydrophilic character given by 4VP to PVC is the most relevant to the SBMA grafting, but 4VP also brings to the system pH-responsive properties. Unfortunately, the SBMA couldn’t graft onto PVC directly by this method since its strong intermolecular attraction avoids interaction with a hydrophobic surface.
- For the manufacturing, I wonder if you bubble the monomer solution in Ar before adding to react with PVC tubes, (not bubbling during the reaction). Would it be helpful?
Answer: We use the Ar bubbled to displace the air in the solution since the oxygen in the air inhibits the polymerization, reducing the grafting. After Ar bubbled, the ampoules were sealed to keep the atmosphere oxygen-free during the reaction. Section 2.3 was modified.
- Important reference should be cited:
|
Langmuir 32 (19), 5019-5028 |
; Langmuir 35 (5), 1642-1651
Answer: Thanks for the suggestion, some of the references were added.

Round 2
Reviewer 1 Report
The quality of the manuscript was strongly improved and the paper can be accepted for publication in its present form.
Reviewer 2 Report
The current version can be published.